# Dissecting the Unique Self-Assembly Landscape of the HIV-2 Capsid Protein

**DOI:** 10.3390/v17101384

**Published:** 2025-10-17

**Authors:** Matthew Cook, Pushpanjali Bhardwaj, Faith Lozano, Christian Freniere, Ryan J. Malonis, Yong Xiong

**Affiliations:** 1Department of Molecular Biophysics and Biochemistry, Yale University, New Haven, CT 06511, USA; matthew.cook@yale.edu (M.C.); na.pushpanjali@yale.edu (P.B.);; 2Current address: Independent Researcher, Madison, WI 53703, USA; 3Current address: Independent Researcher, Houston, TX 77011, USA

**Keywords:** HIV-2, capsid, in vitro assembly, capsid-like particles

## Abstract

Human immunodeficiency virus type 2 (HIV-2) is a lentivirus closely related to HIV-1 but exhibits distinct molecular and clinical features that influence viral infectivity and efficacy of antiretroviral therapy. The HIV capsid is a critical structural component with multifaceted roles during infection and mediates some of the observed divergence between HIV-1 and HIV-2. Unlike HIV-1, study of the HIV-2 capsid is limited and standard protocols for the in vitro assembly of HIV-1 capsid protein (CA) lattice structures have not been successfully translated to the HIV-2 context. This work identifies effective approaches for the assembly of the HIV-2 CA lattice and leverages this to biochemically characterize HIV-2 CA assemblies and mutant phenotypes. Our findings elaborate on the sensitivity of HIV-2 CA to chemical conditions and reveal that it assembles into a more varied spectrum of particle morphologies compared to HIV-1. Utilizing these assemblies, we tested the hypothesis that HIV-1 and HIV-2 employ divergent mechanisms to stabilize CA oligomer forms and investigate the effects of non-conserved substitutions at the CA inter-protomer interfaces. This work advances our understanding of the key biochemical determinants of HIV-2 CA assembly that are distinct from HIV-1 and may contribute to their divergent virological properties.

## 1. Introduction

The human immunodeficiency virus (HIV) capsid is an integral structure involved in interactions with host factors and evasion of immune recognition, shielding the viral genome from the point of cellular entry until its integration into the host genome [1]. Studying the capsid has been a key for improved understanding of HIV biology and has led to an emerging class of highly effective antiretroviral therapeutics [2,3]. HIV consists of two viral lineages, HIV-1 and HIV-2, which arose from distinct simian virus ancestors [4,5,6]. While many characteristics of the capsid are conserved between the two viruses, substantial differences exist that have notable implications on infectivity and host factor interactions. Although HIV-2 is less infectious and progresses less aggressively than HIV-1 [7,8], its infection can nonetheless lead to acquired immunodeficiency syndrome (AIDS) onset and therefore requires lifelong management. The study of HIV-2 will offer benefits both to improving potential treatment for the two million people living with HIV-2 and to deeper insights into the biology of human lentiviruses broadly.

Multiple copies of a single viral capsid protein (CA) self-assemble to form the HIV capsid. The capsid takes the form of a characteristic conical lattice structure composed of approximately 200 CA hexamers and 12 CA pentamers which facilitate high curvature and closure of the viral core [9,10,11,12]. Collectively referred to as capsomeres, both CA oligomeric forms are necessary for infection. The CA capsomeres are stabilized by an inner core of interacting N-terminal domains (NTDs) with a surrounding belt of C-terminal domains (CTDs), as each CA CTD contacts the adjacent protomer NTD [10,12,13,14,15,16,17]. CTD-CTD interactions are responsible for inter-capsomere contacts and are the most readily assembled interface by CA monomers in solution [18,19,20,21,22,23,24]. The propensity of CA to oligomerize has enabled extensive in vitro studies of the HIV-1 capsid using assembled capsid protein. Initially, HIV-1 CA assemblies such as nanotubes or spheres and disulfide-stabilized hexamers were induced to form in buffers containing high salt concentrations (≥1 M NaCl) [13,25,26,27,28]. These platforms provided a variety of means for investigating interactions with viral or host proteins, the effects of CA mutations, and the binding of small molecule inhibitors. The discovery of the stabilizing effect of inositol hexakisphosphate (IP6) enabled the formation of smaller, more conical assemblies at lower pH, which more closely resemble the native capsid [29]. IP6 coordination neutralizes the consecutive rings of positively charged residues lining the central pore of CA capsomeres–a role that is particularly critical for stabilizing CA pentamers, whose central pore is narrower and more densely charged [12,29,30,31,32,33].

The assembled lattice of CA forms interaction surfaces that are frequently better recognized by host factors than monomeric CA [34,35]. Studying these interactions is more readily achieved with reliable methods for introducing assembled CA lattices. This can be especially valuable as many CA functions are mediated by host pro-viral or restriction factors targeting of the CA lattice. Notably, many of these proteins differentially recognize HIV-1 and HIV-2 capsids. For example, MxB appears to restrict HIV-2 infection less potently than HIV-1 [36,37], whereas TRIM5α and NONO more effectively restrict HIV-2 [38,39,40,41,42]. In the case of a pro-viral factor, the host protein cyclophilin A (CypA) interacts differently with HIV-1 and HIV-2 capsids. In HIV-1 infection, CypA binds primarily to the eponymous CypA-binding loop along with two additional sites in the context of assembled HIV-1 CA lattice [43], providing improved evasion of immune sensing and capsid stability [44]. CypA binds HIV-1 CA with sufficient affinity to be incorporated into budding virions and may influence infection as far downstream as the site of genome integration [45,46,47,48]. In contrast, HIV-2 induces no virion packaging of CypA [49,50], exhibits reduced binding affinity for CypA [39,50], and shows only a modest reduction in infectivity upon inhibition of CypA [49,51,52,53]. Despite these differences, single insertion/deletion mutations in the CypA-binding loop are sufficient to swap apparent CypA binding affinities of HIV-1 and HIV-2 [39,52,54].

Whether studying CA assembly properties, host factor interactions, or small molecule inhibitor targeting, in vitro assembly methods of CA have been integral to expanding HIV-1 capsid research. While thoroughly developed in the case of HIV-1, these approaches have not been similarly fruitful for the study of HIV-2 CA. This may be due to the observation that HIV-2 CA is considerably more resistant to induced assembly, with the only reported instance requiring extremely high salt concentrations (≥3 M), in contrast to the conditions typically used for HIV-1 [55]. While this indicates interesting biochemical differences between the two CAs, it limits potential approaches for investigating HIV-2 biology. A more recently developed technique, assembly via templating on functionalized liposomes, has been effective for structural resolution of both HIV-1 and HIV-2 CA assemblies [32,56], but this approach alone is limited with respect to querying biochemical properties. Previous work resolving structures of the domains of HIV-2 CA and of the fully assembled HIV-2 CA lattice have revealed intriguing differences between HIV-1 and HIV-2 [39,56,57], particularly at the interfaces between two protomers [56]. While most of the CA sequence is conserved between HIV-1 and HIV-2, some residues are notably different in chemical character but are also highly conserved within the population of the individual viruses. It has been proposed that some of these residues in HIV-2 may contribute to an alternative mechanism for stabilizing CA in its two capsomere forms; however, this remains to be experimentally tested.

This work reports efficient, robust, and more broadly applicable in vitro assembly approaches for HIV-2 CA. Our research demonstrates increased sensitivity of HIV-2 CA to a number of chemical determinants of assembly, influencing both assembly rate and resultant morphology, with a more expansive range of observed morphologies than HIV-1 CA. We applied these approaches to investigate the effects of CA inter-protomer interface mutations. These results support the previously proposed stabilization mechanisms of HIV-2 CA capsomere forms but reveal some intriguing complexity with respect to the roles of divergent residues between HIV-1 and HIV-2. We anticipate that this study will offer improved tools for investigating diverse lentiviral capsids and provide new context for understanding the assemblies of both HIV-2 and HIV-1.

## 2. Materials and Methods

### 2.1. Protein Expression and Purification

HIV-2 GL-AN CA (GenBank: AAA43932.1) and HIV-1 NL4-3 CA (GenBank: AAK08483.1) genes in pET-11a vectors were used for WT sequences and expression of the respective proteins [58,59]. Point mutations were introduced to the pET-11a HIV-2 GL-AN CA sequence to create all mutants in this paper (Appendix A). The L26V/G39M double mutant of HIV-2 GL-AN CA was cloned into a pET-11a vector containing a C-terminal maltose-binding protein (MBP) and polyhistidine (His) tag. The MBP tag was included to enhance expression efficiency, while the His tag enabled affinity purification. A TEV protease cleavage site was inserted upstream of the tags to facilitate their removal prior to final purification steps. Disulfide-stabilized HIV-1 CA nanotubes were produced using pET-11a HIV-1 NL4-3 CA A14C/E45C-TEV-MBP-6xHis. All constructs were expressed in BL21(DE3) *E. coli* competent cells. Cells were grown to an OD600 of 0.6–0.8 in Terrific broth with 100 μg/mL ampicillin before induction with 500 μM isopropyl β-d-1 thiogalactopyranoside (IPTG) at 18 °C for 16–18 h. Cells were harvested by centrifugation and either used immediately for protein purification or flash frozen and stored at −80 °C.

For purification of untagged proteins, cell pellets were resuspended in lysis buffer (50 mM Tris∙HCl, pH 8.0 at 4 °C; 500 mM NaCl; 0.2 mM tris(2-carboxyethyl)phosphine (TCEP); 1 × Halt™ protease inhibitor cocktail) and lysed in a microfluidizer. The lysate was clarified by centrifugation, and bulk protein was subsequently precipitated by the addition of 35% ammonium sulfate with continuous stirring at 4 °C for 1 h. The precipitated protein was collected via centrifugation, resuspended, and dialyzed overnight into SP_A_ buffer (50 mM Na∙MES, pH 6; 25 mM NaCl; 0.2 mM TCEP). The following day, any insoluble material was removed by centrifugation, and the clarified supernatant was applied to two tandem 5 mL HiTrap SP HP columns (Cytiva, Marlborough, MA, USA) for cation exchange chromatography. The protein was eluted with a gradient from 25 mM NaCl (SP_A_ buffer) to 1 M NaCl (SP_B_ buffer, differing only in NaCl concentration). The WT CA and most mutants bound weakly to the resin and eluted between 10% and 30% through the gradient. These eluted fractions were sufficiently pure for subsequent size exclusion chromatography (SEC). Some mutants (K31A, K170N) did not bind to the resin and the flow through was collected and dialyzed into Q_A_ buffer (identical to SP_A_ but with 50 mM Na∙HEPES, pH 8, instead of Na∙MES). These proteins were then loaded onto a 5 mL HiTrap Q HP column (Cytiva, Marlborough, MA, USA) for anion exchange chromatography, but again did not bind to the resin. The flow-through was concentrated using a 100 kDa MWCO centrifugal filter, allowing the CA protein to pass through while retaining larger contaminants. The resulting filtrate was sufficiently pure for SEC. With mostly pure CA following the ion exchange approaches, the protein was concentrated to an appropriate volume and loaded onto a Superdex™ 75 Prep Grade (Cytiva, Marlborough, MA, USA) 16/600 column (GE) for SEC in appropriate buffer (25 mM K∙HEPES, pH 7; 100 mM potassium glutamate; 0.2 mM TCEP; 5% *v*/*v* glycerol). Purified protein was confirmed by SDS-PAGE and concentrated to between 500 μM and 1.2 mM before being snap frozen and stored at −80 °C.

HIV-2 CA L26V/G39M-TEV-MBP-6xHis and HIV-1 CA A14C/E45C-TEV-MBP-6xHis were purified using similar workflows. Cell pellets were resuspended in lysis buffer (50 mM Tris-HCl, pH 8.0 at 4 °C; 400 mM NaCl; 25 mM imidazole; 0.2 mM TCEP; 1× Halt™ protease inhibitor cocktail) and lysed using a microfluidizer. Lysates were clarified by centrifugation and applied to pre-equilibrated Ni-NTA agarose resin (Qiagen, Hilden, Germany). Following incubation at 4 °C for 30 min, the flow-through was collected, and columns were washed with 30 mL of binding buffer (same as lysis buffer). Proteins were eluted with elution buffer (50 mM Tris-HCl, pH 8.0; 200 mM NaCl; 300 mM imidazole; 0.2 mM TCEP). Eluates were dialyzed overnight at 4 °C into low-salt cleavage buffer (SP_A_ buffer: 25 mM MES, pH 6.0; 25 mM NaCl; 0.2 mM TCEP) in the presence of TEV protease to remove the MBP-6xHis tag. After dialysis overnight, samples were clarified by centrifugation and applied to two tandem HiTrap SP HP columns (Cytiva, Marlborough, MA, USA). Proteins were eluted using a linear gradient to SP_B_ buffer (25 mM MES, pH 6.0; 1 M NaCl; 0.2 mM TCEP). The HIV-2 CA L26V/G39M protein eluted in the flow-through as a single peak confirmed by SDS-PAGE. For HIV-1 CA A14C/E45C, the eluted peak was reloaded onto Ni-NTA resin to further remove residual tagged species. The Ni-NTA column flow-through and wash fractions were collected, confirming acceptable purity by SDS-PAGE. Final samples were concentrated using 10 kDa MWCO centrifugal concentrators (MilliporeSigma, Cork, Ireland) and subjected to SEC on a Superdex™ 75 Prep Grade (Cytiva, Marlborough, MA, USA) 16/600 column (GE, Chicago, IL, USA) equilibrated in SEC buffer (25 mM K·HEPES, pH 7.0; 100 mM potassium glutamate; 0.2 mM TCEP; 5% *v/v* glycerol). Proteins eluted as single peaks, consistent with homogeneity. SDS-PAGE-validated fractions were concentrated, flash-frozen in liquid nitrogen, and stored at −80 °C.

### 2.2. In Vitro Assembly of CA Structures

Initial assemblies were conducted similar to previous reported approaches with HIV-1 CA [25,29], by dialyzing HIV-2 CA at desired concentrations into target buffer solutions at 4 °C. Initial buffers included either 25 mM Tris∙HCl (pH 7.5 at 4 °C) or 25 mM MES (pH 6.0), supplemented with 1 mM IP6 and varying NaCl concentrations (500 mM—3 M). Assembly was monitored (up to 3 days) for changes in turbidity, ability to produce large enough assemblies to be pelleted by tabletop centrifugation, and negative-stain electron microscopy (EM). A similar process was used for inducing assembly and crosslinking of cysteine mutants [13,28]. Assembly was initially induced by dialysis into buffer containing 1 M of either NaCl (HIV-1) or potassium glutamate (HIV-2), 40 mM β-mercaptoethanol (BME), and no IP6 to selectively promote CA nanotube formation, except during the assembly of HIV-2 CA P14C/E45C/W184A/M185A which was assembled in the presence of IP6. This initial dialysis proceeded overnight at 4 °C and the appearance of turbidity was confirmed before transitioning into similar buffer lacking BME, to permit disulfide formation. Dialysis was continued for two days, with the buffer being refreshed once, before a final dialysis into low-salt buffer (150 mM NaCl) for HIV-1 or a range of salt concentrations for HIV-2, as described in Appendix A.

Following confirmation of the assembly benefits of glutamate salts, HIV-2 CA was assembled, unless otherwise noted, by diluting stock protein to 200 μM into final buffer conditions containing 50 mM K∙HEPES (pH 7.0), 1 M potassium glutamate, and 1 mM IP6. This reaction mixture was then incubated at 37 °C for 1 h, although highly assembly-prone reactions often initiated assembly at room temperature (RT) prior to incubation. Some less assembly-prone reactions took several hours to complete at RT or 37 °C. The resultant assemblies remained stable for several days at RT or 4 °C.

### 2.3. Sedimentation of Assembled CA Structures

For sedimentation or co-sedimentation assays, a nominal concentration of 200 μM CA was used for initial assembly (1 h at 37 °C). For co-sedimentation with CypA, CypA was introduced following initial CA assembly incubation and allowed to equilibrate at RT for 30 min. From the total volume of either sedimentation or co-sedimentation assays, an aliquot was taken for pelleting in a tabletop centrifuge, spinning for 10 min at 16,000 rcf at RT. The supernatant was removed and retained. The pellet was washed with buffer matching the original assembly conditions and centrifuged again. This wash supernatant was discarded, and the pellet was resuspended in a volume equivalent to that of the removed supernatant to preserve concentration consistency. Samples were taken from the total (T), supernatant (S), and pellet (P) fractions for SDS-PAGE analysis and negative-stain EM grids were prepared using the total fraction. When attempted with WT HIV-2 CA samples, the pellet sample was not clearly distinguishable from the total sample by negative-stain EM.

### 2.4. Monitoring CA Assembly by Optical Density at 350 nm

For time scale turbidity measurement, 50 μL samples were prepared in a clear, 96-well, flat-bottom plate, with three duplicate sample wells followed by a blank well to which an equivalent volume of water would be added in place of protein. Reactions were initiated via pipetting and gentle mixing of protein (or water for blanks) with a multi-channel pipette into the sample plate. The plate was then loaded into a 37 °C pre-heated SynergyMx (BioTek) plate reader to begin measuring 350 nm absorbance. Absorbance readings were collected from every well every minute for 1 h. With the volume used, additional time carried the risk of evaporation significantly changing the effective concentration in the wells and therefore the behavior of the sample. For analysis, at each time point, the absorbance values of the three sample wells were averaged together and the value of the blank well was subtracted from the result. Standard deviation from the mean was similarly calculated at each time point from the three sample wells before plotting in GraphPad Prism 10.5.0.

### 2.5. Electron Microscopy

For negative-stain EM imaging, sample grids were prepared by applying 3.5 μL of the sample to a glow-discharged (25 mA for 30 s) negative-stain EM grid (carbon on 400-mesh copper; EMS, Hatfield, PA, USA). Sample was allowed to deposit for 1.5 min before blotting with filter paper. The grid was washed with 2% uranyl acetate (UA) stain (EMS, Hatfield, PA, USA) before staining with 2% UA for 1.5 min and blotted until dry. Imaging was performed at the Yale University cryo-EM facility using a 120 kV Talos™ L120C TEM equipped with a CETA CMOS camera.

For cryo-EM imaging, 3.5 μL of sample was applied onto a glow-discharged (15 mA for 45 s) Quantifoil™ R2/1 200-mesh copper grid. The sample grids were dual-side blotted for 6.0 s using a Vitrobot (ThermoFisher, Waltham, MA, USA) under 100% chamber humidity and then plunge-frozen into liquid ethane. Sample screening was conducted at the Yale University cryo-EM facility with a 200 kV Glacios™ cryo-TEM equipped with a K3 direct detection camera. SerialEM was used for capturing images and IMOD was used for visualization [60].

### 2.6. Quantification of CA Assembly Morphologies

Particle morphology was quantified using two strategies in parallel. Following the collection of negative-stain EM micrographs of the CA samples, captured at magnifications sufficient to resolve features such as CA nanotubes and spheres, images were loaded into ImageJ v1.53k for analysis. Image scaling was set using the scale bar built into the micrograph image during data collection. The “polygon selections” tool was then used to circumscribe all particles in a micrograph which could be confidently distinguished and were fully contained within the image. The enclosed shape was then measured for its 2D projection area which was then recorded. Amorphous aggregates were excluded from the analysis, as their structures were not representative of capsid morphology. For each condition, typically more than 50 particles were measured by analyzing all distinguishable particles in each micrograph. A minimum of three micrographs were analyzed per condition, and no additional particles were measured once more than twelve micrographs had been analyzed. GraphPad Prism 10.5.0 was used for statistical comparison and plotting of data. For direct comparisons between two conditions, a Mann–Whitney test was used to compare ranks of the two samples. To evaluate statistical differences between WT HIV-2 CA and multiple mutant samples, a Kruskal–Wallis test was used to compare ranked data and determine significance across groups.

While measuring the 2D projection areas of the imaged particles, each particle was classified as either a sphere, cone, closed nanotube, open nanotube, or sheet. Particles which were not fully contained within the image were excluded, similarly to the prior analysis. Sheets were typically observed in a flattened conformation and were readily distinguishable from other particle types. Spheres were defined as being round and largely uniform. To distinguish spheres from round blobs or aggregates, particles were required to exhibit a clearly defined assembled edge and/or a higher-contrast boundary with a dimmer interior, consistent with other enclosed, assembled particles. Cones and closed nanotubes were distinguished based on aspect ratio and side geometry: closed assemblies with parallel sides exceeding three times their width were classified as nanotubes. Larger structures, whose sides are at angles to one another and therefore produce triangular or heterogenous angular shapes were considered cones. Small pill-shaped structures with slightly elongated axes but parallel sides were also categorized as cones. While most cones and closed nanotubes were easily distinguishable, a small subset at the boundary of classification could reasonably be placed in either category. Closed and open nanotubes were distinguished in that both ends of open nanotubes were clearly not sealed and/or abruptly flat. The same trends in overall morphology would still bear out if open and closed categories were combined into one nanotube category, but we determined that distinguishing them provided useful additional information.

### 2.7. Atomic Model Analysis

Structural analyses were performed using UCSF ChimeraX 1.9 [61,62]. The MatchMaker tool was employed to align individual chains or domains when such alignment facilitated structural comparison [63].

## 3. Results

### 3.1. Unique Self-Assembly Properties of HIV-2 CA Revealed by Robust In Vitro Assembly

Significant advances in the understanding of HIV-1 capsid biology have arisen with the improvement of techniques for the in vitro assembly of CA. However, comparable approaches are largely absent for exploring HIV-2 CA, despite its similarity to HIV-1 CA. We sought to resolve this deficit by identifying efficient and reproducible methods for the in vitro assembly of HIV-2 CA. We purified WT HIV-2 GL-AN CA for testing in this study and therefore this manuscript follows the residue numbering of HIV-2 GL-AN. This numbering is similar to HIV-1 but usually offset by +1 from other HIV-2 strains, including ROD [58]. Given its essential role in stabilizing the HIV-1 capsid, we anticipated that IP6 supplementation would markedly enhance the stability of the HIV-2 CA lattice. However, despite IP6 supplementation, our attempts to assemble the HIV-2 CA lattice using similar approaches to those reported previously failed to produce apparent assembly [25,29,55]. This included failure to reproduce assembly in ≥3 M NaCl, with or without IP6, as evidenced by the absence of product sedimentation upon centrifugation and the lack of ordered assemblies observed by negative-stain EM (Appendix A). These results were perplexing as the general mechanisms underlying HIV-1 or HIV-2 CA assembly were anticipated to be similar.

To explore further, we maintained the overall approach but substituted NaCl with potassium glutamate (KGlu) to induce assembly, reasoning that KGlu had been reported to be a better mimic of physiologic osmolytes and improve in vitro enzymatic efficiency and folding [64,65,66]. Strikingly, KGlu inclusion in HIV-2 CA assembly reactions resulted in rapid increases in turbidity within seconds to minutes, depending on concentration. In contrast to the attempted NaCl-induced assemblies, this turbid material could be pelleted in a tabletop centrifuge and subsequent negative-stain EM imaging revealed cones, spheres, and nanotubes similar to comparable assemblies of HIV-1 CA (Figure 1a–c and Appendix A). Higher resolution imaging by cryo-EM confirmed ordered, lattice-like features of these assemblies (Figure 1d). As with HIV-1 CA, assembly efficiency was assessed by monitoring optical density at 350 nm as a proxy for HIV-2 CA lattice formation [23,67], followed by negative-stain EM imaging to confirm the presence of ordered CA structures. Using this approach, HIV-2 CA appeared to assemble similarly regardless of the cationic component of the salt (sodium or potassium), but the assembly was highly dependent on the anionic component (chloride, acetate, or glutamate) with only glutamate salts consistently producing efficient assembly (Figure 1e). In comparison, while HIV-1 CA lattice also assembled more efficiently in KGlu, it maintained markedly more efficient assembly than HIV-2 CA in NaCl (Figure 1f). Besides the lack of observed assembly in high-NaCl conditions, these results are otherwise consistent with the previously reported characterization that HIV-2 CA assembles less efficiently than HIV-1 CA under similar in vitro conditions [55].

We systematically probed HIV-2 CA assembly across varied IP6, pH, and salt conditions. Consistent with its known role in stabilizing HIV-1 CA assembly and the observation of its incorporation in recent structures of liposome- templated HIV-2 CA assemblies [29,30,56], IP6 concentration correlated positively with HIV-2 CA assembly efficiency. However, IP6 was not required for assembly of HIV-2 CA at high-KGlu concentrations (≥1 M) (Figure 2a,b). Similar to HIV-1, assembly in the absence of IP6 produced almost strictly open-ended HIV-2 CA nanotubes and a significantly higher proportion of sheets (Figure 2c) [25]. This was as anticipated, due to the critical role of IP6 in stabilizing CA pentamer formation and thereby promoting high capsid curvature and closure of CA assemblies [12,17,32,33,56]. However, in contrast to HIV-1, which has been reported to assemble into predominantly conical structures at acidic or neutral pH in the presence of IP6 [17], HIV-2 CA assembled at pH 7.0 formed a wide array of particle morphologies (Figure 1b,c). We therefore tested the role of buffer pH in HIV-2 CA assembly. In the relatively narrow pH range tested (pH 6.0 to 8.0), assembly appeared to proceed most rapidly at lower pH, although all pH conditions resulted in rapidly increased turbidity (Figure 3a,b). Furthermore, negative-stain EM imaging of the products revealed stark differences in particle morphology, following a trend of increasing curvature with decreasing pH (Appendix A). This observation is qualitatively consistent with the reported behavior of HIV-1 CA, which assembles predominantly into conical structures at acidic pH (~6) and tubular structures at basic pH (~8) [17], but the effect appeared to be more pronounced for HIV-2 CA.

To quantitatively characterize the observed particle morphology differences, we used two approaches to analyze the EM image data. In the first approach, we aimed to measure particle size. As twelve CA pentamers are required to close a CA assembly regardless of its size, the difference in size between two closed assemblies informs the ratio of incorporation of hexamers: a small CA sphere may contain only pentamers while a longer, larger enclosed nanotube consists predominantly of hexamers. As such, we determined the area of the 2D projections of CA particles captured by negative-stain EM to serve as a proxy of the size of the particles (Appendix A). In the second approach, we systematically categorized particles as spheres, cones, closed nanotubes, open nanotubes, or sheets to differentiate various morphologies (Appendix A). Using these approaches, we confirmed that lower pH leads to smaller, more curved HIV-2 CA particles, while higher pH produced more tubular assemblies with a transition at around pH 7.0 (Figure 3c). However, the proportion of spherical particles appeared much higher for HIV-2 than has been reported for HIV-1 despite IP6 being present (Figure 3d–g). Most HIV-2 CA particles were classified as spheres below pH 7, while at or above that value, cones and nanotubes predominated, though some spheres remained present. In contrast, HIV-1 CA assembled at pH 7 produced significantly smaller particles (*p* < 0.001) which were mostly conical in shape, rarely forming closed nanotubes or spheres (Appendix A). These characteristics of HIV-1 CA assembly were largely consistent with the literature, though we observed a notable difference in particle size and morphology when HIV-1 CA was assembled in KGlu rather than NaCl (Appendix A). Given that pH 7 represented both a common transition point for particle morphology and a physiologically relevant condition, it was used in all subsequent experiments examining morphological effects.

As expected, CA concentration was strongly correlated with the rate of assembly (Appendix A). However, CA concentration also appeared to influence the morphology of the assembled particles, with higher concentration producing a larger population of cones and lower concentration resulting in a greater proportion of extended tubular assemblies and spheres (Figure 1a,b and Appendix A). At high concentrations, HIV-2 CA assembly more closely resembled that of HIV-1. This is perhaps a result of significant differences in the rate of nucleation, which modeling studies have identified as a major limiting factor for CA assembly [68,69]. Substitutions of engineered cysteines in HIV-1 CA (A14C/E45C or A42C/Q54C, most notably) have been used widely to stabilize HIV-1 CA in vitro assemblies via the designed formation of disulfide bonds [13,28,70]. Comparable residues in HIV-2 CA appeared to be positioned at similar inter-residue distances, prompting our introduction of P14C/E45C or A42C/Q54C mutations. While both mutation pairs were assembly-competent (Appendix A), they did not readily form disulfide bonds to the same degree as HIV-1 CA (Appendix A). Despite this, HIV-2 CA P14C/E45C more readily formed stable nanotubes in high-KGlu concentrations (≥1 M) than WT HIV-2 CA, and these nanotubes remained intact and did not turn into smaller closed structures even when dialyzed into conditions of physiological salinity with IP6 (Appendix A). Due to the large size of the nanotubes, this material provides a more reliable platform for co-sedimentation experiments than the moderately sized HIV-2 CA capsid-like particles (CLPs) templated on liposomes [32,56]. We demonstrated this with a co-sedimentation experiment using cyclophilin A (CypA), which bound appreciably to both nanotubes and lipid-templated CLPs, though the cysteine-stabilized nanotubes consistently retained all detectable CA in the pellet fraction (Appendix A). To test an analogous engineered HIV-1 CA hexamer construct [13], we generated HIV-2 CA P14C/E45C/W184A/M185A, which appeared to spontaneously form hexameric structures upon addition of KGlu and IP6 despite the absence of disulfide formation (Appendix A). These assemblies were substantially enriched following assembly but gradually re-equilibrated into partially disassembled populations over time (Appendix A). This material has not yet been successfully used in a capacity, similar to its HIV-1 CA counterpart, to generate disulfide-crosslinked individual CA hexamers. Further efforts are warranted to engineer more effective stabilizing mutations for HIV-2 CA constructs.

### 3.2. Sequence Divergence at Inter-Protomer Interfaces Modulates CA Assembly Kinetics and Particle Morphology

The HIV-1 and HIV-2 CA sequences are about 80% similar, with many residues at the interface between CA protomers conserved [71]. Nonetheless, assembly differences are clearly observable and likely driven by the small number of divergent residues. To begin understanding the variability in these interfaces, we selected six sets of mutations, targeting ten interfacial residues spanning each of the three protomer-protomer interfaces (Figure 4a and Table 1). These residues were selected due to their divergence in the chemical nature of their sidechains. The selected HIV-2 GL-AN CA sites were mutated to the HIV-1 NL4-3 equivalent residue and resulting proteins were purified. Among these, two constructs consisted of multiple mutations. The mutant L26V/G39M was selected to test the effect of introducing M39, a residue implicated in capsomere stability but previously shown to disrupt HIV-2 core assembly [16,17,57]. In this construct, the L26V revertant mutation was included to mitigate potential steric clash with M39 [56]. The mutant D178S/P179Q/A180E was introduced because the neighboring residues likely influence one another structurally. Specifically, S178 and E180 are known to interact in HIV-1 CA based on structural and biochemical studies [18,19,20,72]. Meanwhile, P179 is divergent in HIV-2 but may also alter the local loop geometry, potentially disrupting the interaction between residues S178 and E180. To account for this context-dependent structural interplay, the entire contiguous stretch was mutated as a unit.

Previous work has described the assembly of CA molecules into a capsid as a finely tuned system to maintain core integrity for immune evasion but not inhibit genome release for integration [73]. Given this, we hypothesized that among the set of substituted interface mutants, some would enhance the rate of assembly while others would inhibit it. Conducting assembly assays by monitoring turbidity as described previously, we confirmed this expectation and could separate the mutations by their rate of assembly compared to WT (Figure 4b,c and Appendix A). Pelleting of assembled products and negative-stain EM were also used to support these results (Appendix A). The Q41S and Y50Q mutants exhibited markedly slower assembly rates. Both mutations are at the NTD-NTD interface and were expected to feature increased contacts in the native WT HIV-2 CA context, therefore the reduced efficiency of these mutations appeared consistent with the structurally observed interactions. In the case of Q41S, the observed turbidity appeared to arise predominantly from amorphous aggregation rather than the formation of ordered assemblies (Appendix A). Assembly reactions with Q41S were also highly variable and, in some instances, failed to produce any higher-order structures (Appendix A). In contrast, the mutants L26V/G39M, K31A, D61G, and D178S/P179Q/A180E exhibited an increased rate of assembly and achieved higher final turbidity. L26V/G39M at the NTD/NTD interface was anticipated to introduce additional surface area for hydrophobic contact while avoiding potential steric clash [56]. Consistent with this, the mutant assembled too quickly to measure its rate with the approach described, but producing the greatest turbidity following assembly (Appendix A). The CA L26V/G39M particles were consistently well-ordered as inspected by negative-stain EM (Appendix A). D178S/P179Q/A180E is present at the CTD-CTD interface and was expected to provide increased hydrogen bonding between capsomeres, an interpretation which also appeared consistent with the results. K31A and D61G are located at the NTD–CTD junction and appeared to participate in inter-protomer interactions exclusively in HIV-2 structures. As such, the K31A and D61G mutations were expected to disrupt these contacts. The observation that both mutants exhibited enhanced assembly kinetics was therefore unexpected, but may be a consequence of the increased flexibility of the mutated residues. The HIV-2 CA mutants which exhibited greater assembly efficiency than WT were further tested for assembly competency in NaCl. Only the L26V/G39M mutant was able to assemble efficiently in the presence of NaCl (Appendix A), whereas mutants K31A, D61G, and D178S/P179Q/A180E showed little to no assembly, similar to WT (Appendix A). Even for L26V/G39M, increasing NaCl concentration inhibited assembly and the structures that formed at high salt were predominantly tubular (Appendix A).

Several of these substituted residues were hypothesized to play roles in managing the pleomorphism of HIV-2 CA [56]. Some appeared to form interactions within both hexameric and pentameric capsomeres while others formed inter-protomer contacts in only one form. To further test the hypothesis that these residues are mechanistically important for HIV-2 CA pleomorphism, we introduced mutations to disrupt only one of the capsomere forms (Figure 5a). The Q41E/N57D mutant was designed to disrupt pentamer formation by replacing a stabilizing hydrogen bond with a repulsive charge interaction. This change was expected to mainly affect the pentameric form, while preserving a polar interaction between Q41E and Y50 in the hexamer. Conversely, Y50 and K170 both appeared to interact with the adjacent protomer in the hexameric form but were not involved in the pentameric structure. Therefore, Y50Q/K170N was expected to disrupt hexamer formation. Since pentamers are essential for introducing curvature and closing CA assemblies, structures with a high ratio of pentamers are expected to be more curved and smaller, whereas those with a high proportion of hexamers are likely to be flatter and larger. Both mutants were purified, assembled, and analyzed by negative-stain EM (Figure 5b,c). Quantification of these results as described previously revealed stark shifts in the morphology of the CA particles consistent with the predicted effect on capsomere stabilities (Figure 5d,e). Assemblies of CA Q41E/N57D were significantly larger than WT (*p* < 0.0001) and exhibited much greater proportions of tubular assemblies including mostly open-ended nanotubes, supporting the notion of low incorporation of pentamers. Conversely, CA Y50Q/K170N particles were extremely small (*p* < 0.0001) and round, almost exclusively forming spheres, indicating low incorporation of hexamers. Following similar rationales, the individual mutations Y50Q and K170N were expected to destabilize hexamer formation. Assemblies of CA Y50Q were indeed significantly smaller than WT assemblies (*p* < 0.001), though K170N assemblies were not significantly different (*p* > 0.05; Figure 5d,e and Appendix A). Another mutation, K170A produced a more significant phenotype than K170N, but assembled less effectively, frequently forming amorphous structures (Appendix A). This aberrant assembly behavior was consistent with previously reported properties of HIV-1 CA K170A constructs [73]. Both Y50Q and K170N produced mostly conical constructs with only occasional closed nanotubes (Figure 5e and Appendix A). Overall, these data were consistent with the individual mutations Y50Q and K170N modestly destabilizing hexamer formation, while the combined mutant Y50Q/K170N produced an additive effect on hexamer destabilization.

In addition to the designed disruption mutations, we also analyzed the substitutions to HIV-1 residues to assess their effect on HIV-2 assembly morphology. HIV-2 CA mutants were assembled and imaged by negative-stain EM with subsequent quantification of particle morphology (Figure 5d,e and Appendix A). Mutants D61G and D178S/P179Q/A180E did not produce a significant difference in the mean size of particles compared to WT, though by morphology classification, D178S/P179Q/A180E exhibited greater proportions of conical assemblies with fewer spheres and nanotubes, similar to HIV-1. CA L26V/G39M, K31A, and Y50Q (also described above) produced particles which were significantly smaller than WT and predominantly conical. Q41S formed mostly amorphous aggregates, but the ordered assemblies which could be identified were generally small and spherical. Overall, the HIV-1 mimicking mutations showed a reduced tendency to form tubular structures and a greater propensity to assemble into cones, similar to the differences observed between WT HIV-1 and HIV-2 CA molecules.

## 4. Discussion

While in vitro assembly processes likely differ from native capsid maturation, they provide a robust platform for studying capsid protein biochemistry and interactions with host factors. This approach has proven effective for investigating the HIV-1 capsid, and we have now successfully adapted it for application to HIV-2. The conditions we use to induce HIV-2 CA assembly are reliable and repeatable, but still sensitive to small changes in the chemical environment or the protein variant. Even more so than HIV-1 CA, HIV-2 CA appears starkly sensitive to the salt and pH conditions during assembly, with large effects on rate of assembly and resulting morphology. HIV-2 CA assemblies were more likely to form into spheres or sheets than has been previously reported of HIV-1 CLPs. Contrary to its role in promoting HIV-1 CA in vitro assembly, the presence of NaCl appeared to inhibit HIV-2 CA assembly. The mechanisms underlying this could be of further interest, both to better understand the evolutionary trajectories or pressures shaping lentiviral capsids and to inform future strategies for pharmacological targeting of the capsid. More generally, the consistent improvement in assembly efficiency for both HIV-1 and HIV-2 CA with glutamate salts may offer noticeable potential as a broader strategy for the in vitro assembly of other primate and non-primate lentiviral capsid proteins.

Testing mutations at the inter-protomer interfaces supported the idea that capsid assembly is finely tuned, with some mutations enhancing assembly efficiency and others impairing it. The substantial disruptions in assembly efficiency by the Y50Q and the Q41S mutations supported the previously discussed hypothesis that these unique contacts in HIV-2 play important roles for stabilizing the HIV-2 capsid [56]. Of note, Y50 appears to be the ancestral residue identity and Q50Y revertant mutants of HIV-1 sensitize the virus to TRIM restriction [74]. Despite Q41 forming inter-protomer contacts in both HIV-2 capsomeres, only small, round CA Q41S assemblies were observed. This may indicate that Q41 is more important for stabilizing the HIV-2 CA hexamer than the pentamer. Consistent with structural predictions, the D178S/P179Q/A180E mutant promoted greater rate of CA assembly. However, the lack of a significant change in particle size may reflect the spatial separation of these residues from intra-capsomere interfaces. Similarly, the L26V/G39M mutant assembled much more readily, providing further support for the hypothesis that M39 performs a significant stabilizing role ancestrally and in most modern primate lentiviruses except for the HIV-2 lineage and its close relatives, which appear to have evolved compensatory mutations that allow tolerance of the otherwise destabilizing M39G substitution [56]. The emergence of L26V may have helped avoid the aberrant assembly phenotype described for HIV-2 ROD M38G [56,57]. On the other hand, K31A and D61G were intriguing in that they significantly increased the rate of assembly despite replacing two interacting residues in the HIV-2 structures with non-interacting ones. A possible explanation is increased flexibility from the two substituted residues, with G61 especially important as part of the HIV-1 TVGG loop implicated in the hexamer–pentamer switch [16,17]. It is possible that these more flexible residues permit a lower-energy transition and/or final structure compared to their HIV-2 counterparts, although this hypothesis would need substantial follow-up studies to test. Regardless, these results offer additional insights into the high degree of conservation of these residues in HIV-1.

While not all the interface substitutions were statistically different in mean particle sizes, it was an overall trend that the introduction of HIV-1 residues produced smaller and more frequently conical assemblies. This trend is consistent with the intrinsic assembly differences in WT HIV-1 CA compared to HIV-2 CA. One interpretation of the overall difference in particle size and increased prevalence of nanotubes in HIV-2 compared to HIV-1 or HIV-1-mimicking mutants is that hexamers are incorporated at a higher ratio in HIV-2. Complicating this simple interpretation, CA spheres also were produced much more readily in HIV-2 than in HIV-1, suggesting pentamers are also relatively stable in HIV-2. Assembly rate likely also influenced final morphology, as a higher rate of assembly for either HIV-1 or HIV-2 CA tended to produce more conical assemblies. CA sheets were primarily observed in lower assembly rate reactions. However, among the HIV-1-mimicking mutants of HIV-2 CA, L26V/G39M and D61G both assembled at higher rates but had distinct effects on particle morphology. Some of these discrepancies may be related to differing rates of nucleation or the stability of the trimers-of-dimers that are expected to mediate self-assembly [75,76,77]. While the current findings remain inconclusive on this point, they present an intriguing case for further study and modeling to clarify the nature of these differences.

An additional rationale for examining the divergent residues at the CA inter-protomer interfaces stems from prior evidence of strong selective pressure acting on HIV CA [78,79], raising the possibility that their divergence reflects adaptive evolution rather than genetic drift. As the stability of the capsid and its capacity for timely disassembly are essential to the replication of the virus, significant penalties are likely incurred by disrupting this balance. Mutations that either stabilize or destabilize CA assembly may confer additional fitness benefits, such as evasion of restriction factors or enhancing interactions with pro-viral factors. Other compensatory interface mutations may enable retention of these additional fitness advantages by offsetting the deleterious change in capsid stability. With this rationale in mind, divergent residues which affect capsid stability or morphology may warrant special attention when investigating distinctions between HIV-1 and HIV-2. A notable example is HIV-2 CA Y50 (Y49 in ROD numbering), which has been implicated in TRIM5 recognition and NONO binding [42,74].

Relatedly, a primary application of in vitro CA assemblies is the characterization of host factors that interact with the capsid. The previously described liposome-templating approach offers a promising strategy for structural analysis, while untemplated assemblies serve as a complementary platform for reductionist biochemical studies. The readily observed binding between CypA and HIV-2 CA lattice assemblies may appear somewhat surprising given the previously reported low binding affinity between CypA and HIV-2 CA [39], the presumed rigidity of the HIV-2 CypA-binding loop [74], and structural modeling suggesting potential steric clashes between bound CypA and adjacent CA chains [56]. Further elucidating this interaction, along with related host factors such as Nup358/RanBP2 [53,80,81], in the HIV-2 background may yield important insights into the distinct host engagement strategies. While the CA assemblies described here were largely robust, additional confidence would be gained by follow-up work to design disulfide-crosslinked capsomeres mirroring those that have been effective in HIV-1 study. The P14C/E45C pair did appear to produce more stable HIV-2 assemblies, but other options—such as engineered disulfide bridges spanning the NTD-CTD interface within individual capsomeres or the CTD-CTD interface to covalently stabilize larger assemblies—may be promising future avenues [82,83].

Overall, we expect that the results described here will contribute both to improved understanding of HIV-2 biology as well as broader lentiviral biology. The development of improved tools for biochemical studies with HIV-2 CA enables broader investigation across a range of research topics. Capsid targeting by both pro-viral and restrictive host factors represents an important but underexplored area of research in HIV-2 biology, especially given the apparent divergence in these interactions between HIV-1 and HIV-2. Further investigation could therefore yield insights relevant to both viruses. Moreover, the observed assembly differences, along with the diverse effects of the interface mutants, provide additional motivation to include HIV-2 into ongoing investigations of capsid maturation and in vivo assembly dynamics. Finally, and perhaps at greatest benefit to individuals living with HIV-2, studies regarding the efficacy of current capsid inhibitors or design of new ones intended for HIV-2 should be more achievable as we expand the array of biochemical techniques for HIV-2 CA research.

### Limitations of the Study

We would note that there are limitations to the analytical approaches we have employed in this work. With respect to the assays monitoring turbidity or using sedimentation as proxies for assembly, these methods are not sensitive to large structures arising from disorganized, amorphous assembly as opposed to the ordered, capsid lattice structures of interest. We have attempted to ameliorate this by checking the assembled products by negative-stain EM and have reported the results as such. Relatedly, our methods for providing quantitative metrics for our negative-stain EM imaging are limited by the non-random nature of particle distribution or deposition on EM grids. While we attempted to account for this by quantifying many images and from multiple grid positions, we caution against over-interpretation of small differences. The analysis of particle area is not sensitive to lattices which are disrupted due to defects or breaks, restricting specific calculation of pentamer-to-hexamer ratios. This analysis is instead intended to systematically describe the trends in particle size which would be demonstrative of changes to this ratio. This study only uses the CA sequences of HIV-2 GL-AN and HIV-1 NL4-3. Both were selected as commonly used strains in laboratory settings, but they do not fully represent the diversity of sequences present for either virus.

## Figures and Tables

**Figure 1 viruses-17-01384-f001:**
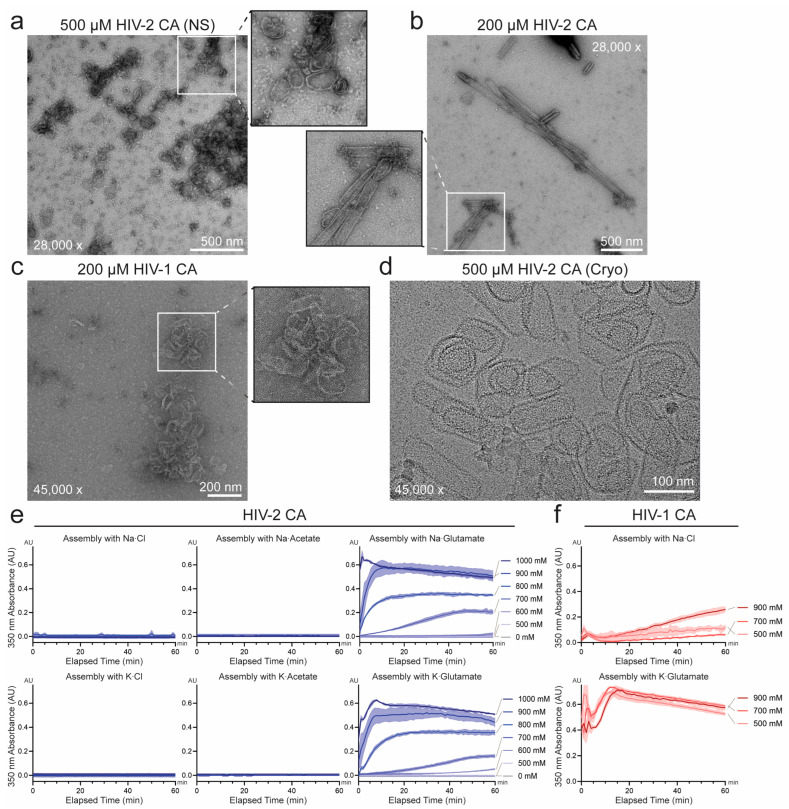
In vitro assembly of HIV-2 capsid protein (CA) into capsid-like particles. (**a**) and (**b**) Negative-stain electron microscopy (EM) micrographs of in vitro assembled HIV-2 CA at 500 μM (**a**) and 200 μM (**b**) concentrations in the presence of 1 M potassium glutamate (KGlu). (**c**) In vitro assembly of HIV-1 CA (200 μM) under identical assembly conditions to (**b**) (1 M KGlu) visualized by negative-stain EM. (**d**) Representative cryo-EM micrograph of WT HIV-2 CA (500 μM) assembled in the presence of 2.5 mM inositol hexakisphosphate (IP6), and 1 M KGlu. (**e**) Assembly kinetics of HIV-2 CA (200 μM) monitored by light scattering at 350 nm, with varying salts and concentrations. Solid lines connect the mean values of adjacent time points (*n* = 3) and shaded area denotes standard deviation from the mean. (**f**) Assembly kinetics of HIV-1 CA (200 μM) under varying concentrations of sodium chloride or potassium glutamate as described in (**e**).

**Figure 2 viruses-17-01384-f002:**
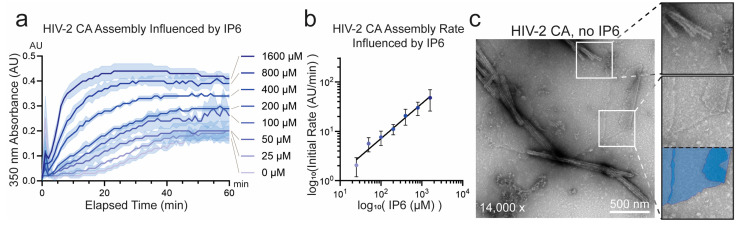
Characterization of IP6 conditions affecting the in vitro assembly of HIV-2 CA. (**a**) Assembly of HIV-2 CA monitored by absorbance at 350 nm at varying concentrations of IP6. The solid line connects the mean values of adjacent time points (*n* = 3) and the shaded area represents the standard deviation from the mean. (**b**) Plot of the initial assembly rate of HIV-2 CA as a function of IP6 concentration, using a logarithmic scale for the rate axis. The initial rate was determined by calculating the slope of a linear regression line fitted to the approximately linear initial phase of assembly; error bars represent the standard error of the estimated slope. The marked line is the linear regression of the logarithmically transformed initial rates as a function of IP6 concentration (*R*^2^ = 0.98). (**c**) Representative negative-stain EM micrograph of WT HIV-2 CA assembled in the absence of IP6, demonstrating the formation of nanotubes and sheets (refer to Figure 1b for assembly with high IP6 concentration). Lower inset illustrates sheet segments shaded blue.

**Figure 3 viruses-17-01384-f003:**
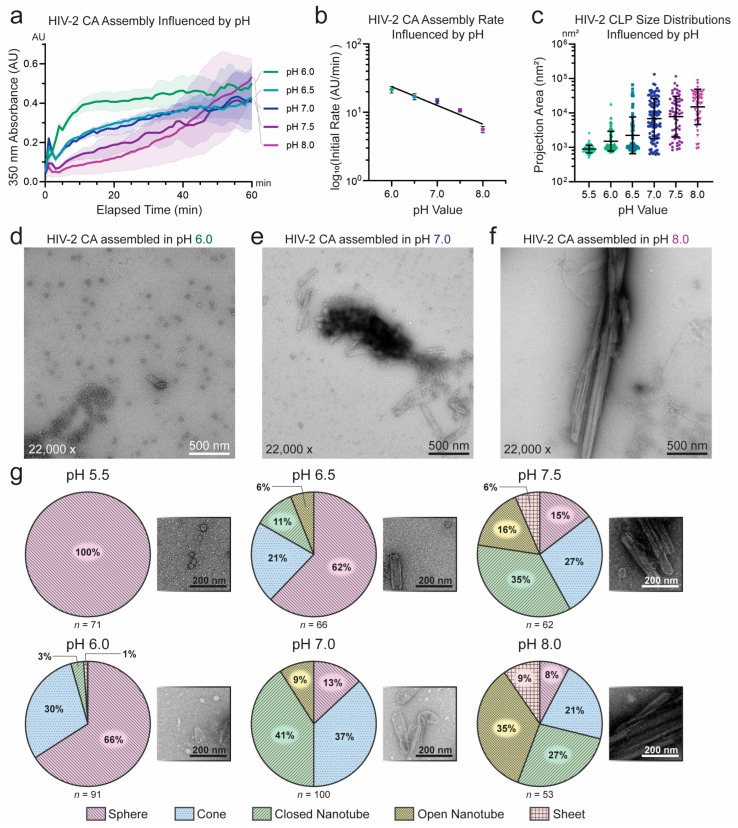
Characterization of pH conditions influencing the in vitro assembly of HIV-2 CA. (**a**) Assembly of HIV-2 CA measured by absorbance at 350 nm with varying pH. Solid line connects mean values of adjacent time points (*n* = 3). Shaded area denotes the standard deviation from the mean. (**b**) Plot of the initial assembly rate of HIV-2 CA as a function of buffer pH, using a logarithmic axis for assembly rate. The initial rate was determined by fitting a linear regression to the approximately linear phase of assembly; error bars denote the standard error of the estimated slope. The marked line is the linear regression of the logarithmic transformation of the slopes as a function of pH (*R*^2^ = 0.92). (**c**) Size of in vitro assemblies of HIV-2 CA across varying buffer pH conditions, measured by 2D projection area on a logarithmic axis. The central line indicates the geometric mean, while the upper and lower lines represent the geometric standard deviations from the mean. Representative negative-stain EM micrographs show HIV-2 CA assembled at pH 6.0 (**d**), pH 7.0 (**e**), and pH 8.0 (**f**). (**g**) Proportions of in vitro assemblies of HIV-2 CA influenced by pH, classified by particle morphology, with representative fragments from negative-stain EM micrographs. Representative whole micrographs for all pH conditions are presented in Appendix A.

**Figure 4 viruses-17-01384-f004:**
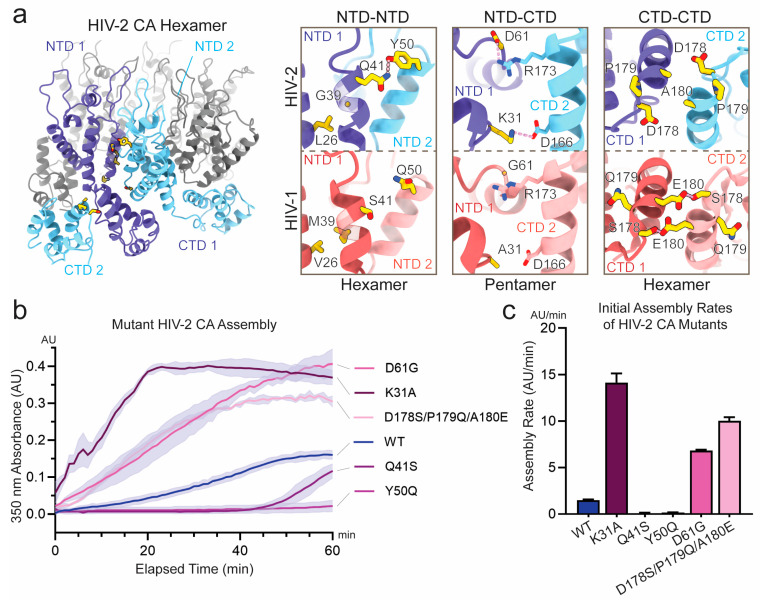
Assembly properties of HIV-2 CA mutants engineered to resemble HIV-1. (**a**) Atomic models highlighting divergent residues at inter-protomer interface contact points. Divergent residues are shown in gold and modeled ionic or hydrogen bond interactions are indicated in pink. Left: hexameric assembly of HIV-2 CA (PDB: 9CNS [56]) providing global orientation for the interfaces of interest. Left box: hexameric NTD-NTD interface (top, HIV-2 PDB: 9CNS [56]; bottom, HIV-1 PDB: 8CKV [16]). Middle box: pentameric NTD-CTD interface (top, HIV-2 PDB: 9CNT [56]; bottom, HIV-1 PDB: 8CKW [16]). Right box: hexameric CTD-CTD interface (PDB entries the same as in left box). (**b**) Assembly of HIV-1 chimeric HIV-2 CA mutants monitored by absorbance at 350 nm in the presence of 700 mM KGlu. The line connects mean values of adjacent time points (*n* = 3), and the shaded area denotes the standard deviation from the mean. (**c**) Histogram of initial assembly rates of HIV-2 CA mutants in the presence of 700 mM KGlu. Initial rates were determined by calculating the slope of a linear regression line fitted to the approximately linear initial phase of assembly; error bars denote the standard error of the estimated slope.

**Figure 5 viruses-17-01384-f005:**
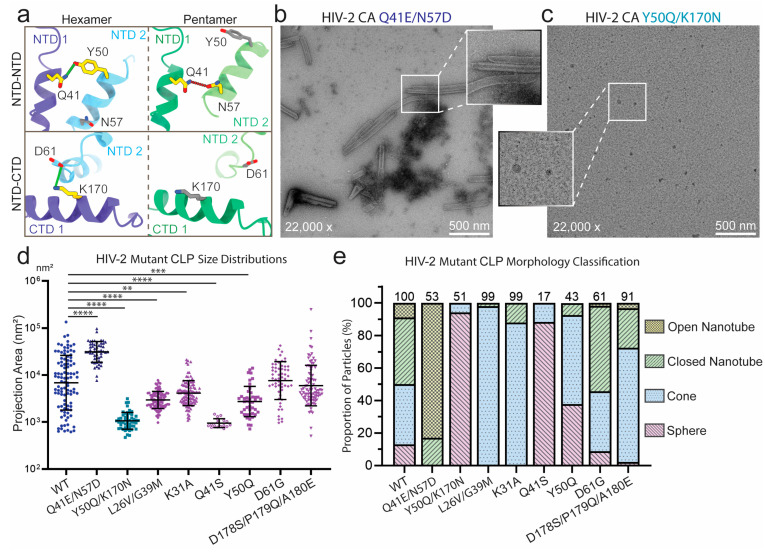
Role of divergent CA residues in determining particle morphology. (**a**) Atomic models of HIV-2 CA in hexameric (left; PDB: 9CNS [56]) and pentameric (right; PDB: 9CNT [56]) conformations, emphasizing inter-protomer contacts targeted for disruption by capsomere-destabilizing mutations. Representative negative-stain EM micrographs of in vitro assemblies of HIV-2 CA mutants designed to disrupt specific capsomere interfaces: (**b**) CA Q41E/N57D primarily formed tubular assemblies, although closed ends were still observed. (**c**) CA Y50Q/K170N formed spherical assemblies infrequently. (**d**) Sizes of the in vitro assemblies of HIV-2 CA WT, Q41E/N57D, Y50Q/K170N, and HIV-1-like mutations measured by 2D projection area on a logarithmic axis. Central line marks the geometric mean with upper and lower lines marking the geometric standard deviations from the mean (Significance values: ** = *p* < 0.01; *** = *p* < 0.001; **** = *p* < 0.0001). (**e**) Proportions of in vitro assemblies of HIV-2 CA WT, capsomere-disrupting mutants, and HIV-1-like mutants classified by particle morphology. Particle counts are indicated above each bar.

**Table 1 viruses-17-01384-t001:** Divergent residues selected for targeted mutagenesis of HIV-2 CA.

Position Number	Residues with >10% Frequency		CA Inter-Protomer Interface Engaged
	HIV-2	HIV-1	
26	L	V	NTD-NTD
31	K	A/G	NTD-CTD, pentamer only
39	G	M	NTD-NTD
41	Q	S/T	NTD-NTD
50	Y	Q	NTD-NTD, hexamer only
61	D	G	NTD-CTD
178	D	T/S	CTD-CTD
179	P	Q	N/A
180	A	E/D	CTD-CTD

## Data Availability

The plasmids described in this study are available upon request. Any data or additional information required for re-analysis is also available upon request. Please direct requests to the corresponding author, Y.X.

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
