# Peer review of "Dissecting the Unique Self-Assembly Landscape of the HIV-2 Capsid Protein"

_viruses, 2025, doi:10.3390/v17101384_

Round 1

Reviewer 1 Report

Comments and Suggestions for Authors

In this manuscript, Cook et al describe new methods for investigating the assembly characteristics of the HIV-2 CA protein. Such methods are well described and used frequently for HIV-1 CA. This manuscript applies what is known about HIV-1 CA assembly to develop assays that will facilitate a better understanding of HIV-2 biology, an understudied and often unappreciated sub-field of basic HIV research. This is a worthy endeavor that has been carried out carefully and rigorously by the authors. It is an important contribution to the field that will hopefully seed further studies. This reviewer is appreciative of the time and effort that the authors directed toward this important work. I have only minor suggestions for the authors to consider.

Page 7 Line 311: I think “Figure 1d: should be “Figure 1C”. Though there is some lack of clarity about whether the Figure 1d panel is a cryo-EM (text) or negative-stain EM (fig legend) image.

Page 11 Lines 401-404: “Despite this, HIV-2 CA P14C/E45C more readily formed stable nanotubes” – Please specify what is being compared. I assume it is WT HIV-2 CA, but was a little confused when reading this sentence.

Page 12 Line 468: “Figure S6c” should be “S6d”   

Page 14 line 528: “Figure 5d,f” should be “5d,e”. There is no panel “f”

Page 9: This is a minor comment that I think the authors should consider about their presentation. The two paragraphs on this page seem to imply that pH is the main driver of tube vs cone assembly. This is mentioned the first sentence of the first paragraph with a reference to 2001 paper (before IP6 was used in these assays) that interpreted spherical HIV-1 CA assemblies at pH<7 as immature-like lattices. I looked up the paper and there aren’t many images from which this interpretation can be evaluated. Either way I think a more recent paper should probably be cited when discussing tube vs cone assembly. Additionally, my understanding is that the presence of IP6 is a much important driver of the assembly of HIV-1 CA into cones vs. tubes. This is mentioned in the middle of the second paragraph, but it comes across as an almost ancillary point relative to the effect of pH. I might consider starting with IP6 and introducing the effects of pH afterward.

Author Response

Comments 1: In this manuscript, Cook et al describe new methods for investigating the assembly characteristics of the HIV-2 CA protein. Such methods are well described and used frequently for HIV-1 CA. This manuscript applies what is known about HIV-1 CA assembly to develop assays that will facilitate a better understanding of HIV-2 biology, an understudied and often unappreciated sub-field of basic HIV research. This is a worthy endeavor that has been carried out carefully and rigorously by the authors. It is an important contribution to the field that will hopefully seed further studies. This reviewer is appreciative of the time and effort that the authors directed toward this important work. I have only minor suggestions for the authors to consider.

Response 1: We thank the reviewer for the encouraging comments and taking the time to review our manuscript.

Comments 2: Page 7 Line 311: I think “Figure 1d: should be “Figure 1C”. Though there is some lack of clarity about whether the Figure 1d panel is a cryo-EM (text) or negative-stain EM (fig legend) image.

Response 2: We thank the reviewer for noting this and we agree that this figure reference is in error. We have modified Figure 1 to switch the previous 1c and 1d panels which had been intended in the text, but had not been changed in the figure itself nor the associated legend. As such, the legend descriptions for 1c and 1d have also been switched to match (Page 8, lines 325-328).

Comments 3: Page 11 Lines 401-404: “Despite this, HIV-2 CA P14C/E45C more readily formed stable nanotubes” – Please specify what is being compared. I assume it is WT HIV-2 CA, but was a little confused when reading this sentence.

Response 3: We agree that this would be better clarified and have amended the sentence to read: “Despite this, HIV-2 CA P14C/E45C more readily formed stable nanotubes in high KGlu concentrations (≥ 1 M) than WT HIV-2 CA…” (Page 11, lines 407-409).

Comments 4: Page 12 Line 468: “Figure S6c” should be “S6d”   

Response 4: We thank the reviewer for noting this, the proper reference should indeed be S6d. We have amended the text accordingly (Page 12, line 476).

Comments 5: Page 14 line 528: “Figure 5d,f” should be “5d,e”. There is no panel “f”

Response 5: We thank the reviewer for noting this, the proper reference should indeed be 5d,e. Further, while reviewing this segment we determined that the accompanying supplemental figure reference should be amended from S8a-c to S8a,b. The figure reference has been amended to: “Figures 5d,e and S8a,b” (Page 15, line 550).

Comments 6: Page 9: This is a minor comment that I think the authors should consider about their presentation. The two paragraphs on this page seem to imply that pH is the main driver of tube vs cone assembly. This is mentioned the first sentence of the first paragraph with a reference to 2001 paper (before IP6 was used in these assays) that interpreted spherical HIV-1 CA assemblies at pH<7 as immature-like lattices. I looked up the paper and there aren’t many images from which this interpretation can be evaluated. Either way I think a more recent paper should probably be cited when discussing tube vs cone assembly. Additionally, my understanding is that the presence of IP6 is a much important driver of the assembly of HIV-1 CA into cones vs. tubes. This is mentioned in the middle of the second paragraph, but it comes across as an almost ancillary point relative to the effect of pH. I might consider starting with IP6 and introducing the effects of pH afterward.

Response 6: We thank the reviewer for this suggestion and agree that other readers would find that to be a more helpful path into discussing the data. Therefore, we have amended the order of introduction of these sets of data, alternating the listings of Figures 2 and 3, and adjusted some of the transitionary text to reflect this. We now cite Schirra et al., 2023 as a more recent study incorporating IP6 into in vitro CA assembly and reporting the bifurcation into curved and tubular assembly products (citation 17). As such, the text now reads as follows (bold + italicized new text, Pages 9-10, lines 335-403):

We systematically probed HIV-2 CA assembly across varied IP6, pH, and salt conditions. Consistent with its known role in stabilizing HIV-1 CA assembly and the observation of its incorporation in recent structures of liposome- templated HIV-2 CA assemblies [29, 30, 56], IP6 concentration correlated positively with HIV-2 CA assembly efficiency. However, IP6 was not required for assembly of HIV-2 CA at high KGlu concentrations (≥1 M) (Figure 2a,b). Similar to HIV-1, assembly in the absence of IP6 produced almost strictly open-ended HIV-2 CA nanotubes and a significantly higher proportion of sheets (Figure 2c) [25]. This was as anticipated, due to the critical role of IP6 in stabilizing CA pentamer formation and thereby promoting high capsid curvature and closure of CA assemblies [12, 17, 32, 33, 56]. However, in contrast to HIV-1, which has been reported to assemble into predominantly conical structures at acidic or neutral pH in the presence of IP6 [17], HIV-2 CA assembled at pH 7.0 formed a wide array of particle morphologies (Figure 1b,c). We therefore tested the role of buffer pH in HIV-2 CA assembly. In the relatively narrow pH range tested (pH 6.0 to 8.0), assembly appeared to proceed most rapidly at lower pH, although all pH conditions resulted in rapidly increased turbidity (Figure 3a,b). Furthermore, negative-stain EM imaging of the products revealed stark differences in particle morphology, following a trend of increasing curvature with decreasing pH (Figure S2). This observation is qualitatively consistent with the reported behavior of HIV-1 CA, which assembles predominantly into conical structures at acidic pH (~6) and tubular structures at basic pH (~8) [17], but the effect appeared to be more pronounced for HIV-2 CA.

*Figure 2 and legend – previously labeled as Figure 3*

To quantitatively characterize the observed particle morphology differences, we used two approaches to analyze the EM image data. In the first approach, we aimed to measure particle size. As twelve CA pentamers are required to close a CA assembly regardless of its size, the difference in size between two closed assemblies informs the ratio of incorporation of hexamers: a small CA sphere may contain only pentamers while a longer, larger enclosed nanotube consists predominantly of hexamers. As such, we determined the area of the 2D projections of CA particles captured by negative-stain EM to serve as a proxy of the size of the particles (Figure S3a,b). In the second approach, we systematically categorized particles as spheres, cones, closed nanotubes, open nanotubes, or sheets to differentiate various morphologies (Figure S3a,c). Using these approaches, we confirmed that lower pH leads to smaller, more curved HIV-2 CA particles, while higher pH produced more tubular assemblies with a transition at around pH 7.0 (Figure 3c). However, the proportion of spherical particles appeared much higher for HIV-2 than has been reported for HIV-1 despite IP6 being present (Figure 3d-g). Most HIV-2 CA particles were classified as spheres below pH 7, while at or above that value, cones and nanotubes predominated, though some spheres remained present. In contrast, HIV-1 CA assembled at pH 7 produced significantly smaller particles (p < 0.001) which were mostly conical in shape, rarely forming closed nanotubes or spheres (Figure S4a,b). These characteristics of HIV-1 CA assembly were largely consistent with the literature, though we observed a notable difference in particle size and morphology when HIV-1 CA was assembled in KGlu rather than NaCl (Figure S4a,c). Given that pH 7 represented both a common transition point for particle morphology and a physiologically relevant condition, it was used in all subsequent experiments examining morphological effects.

As expected, CA concentration was strongly correlated with the rate of assembly (Figure S4c). However, CA concentration also appeared to influence the morphology of the assembled particles, with higher concentration producing a larger population of cones and lower concentration resulting in a greater proportion of extended tubular assemblies and spheres (Figures 1a,b and S4d,e). At high concentrations, HIV-2 CA assembly more closely resembled that of HIV-1. This is perhaps a result of significant differences in the rate of nucleation, which modeling studies have identified as a major limiting factor for CA assembly [68, 69]. Substitutions of engineered cysteines in HIV-1 CA (A14C/E45C or A42C/Q54C, most notably) have been used widely to stabilize HIV-1 CA in vitro assemblies via the designed formation of disulfide bonds…”

Reviewer 2 Report

Comments and Suggestions for Authors

Cook et al evaluate the assembly kinetics and biochemical characteristics of HIV-2 capsid protomers compared to what it known regarding HIV-1 assembly requirements. While much is known with regard to HIV-1, HIV-2 is much less studied in all capacities. The authors thoroughly and carefully investigate numerous experimental conditions that impact HIV-2 capsid assembly positively or negatively with respect to “normal” conditions established for HIV-1 capsid. Additionally, the authors investigate the functional contribution of divergent HIV-1 and HIV-2 amino acid pairs between CA-CA interfaces and the impact on VLP assembly. Overall, this is a well-written and thorough study with very nice complementary data provided as supplemental material. I have no major concerns regarding this study and only have minor comments/questions.

  • Is the HIV-2 capsid sensitive to Lenacapavir?

  • It would be interesting to determine if the HIV-2 CA-CA “revertant” mutants that exhibit enhanced capsid assembly kinetics also exhibit increase infection efficiency compared to WT or mutants that assemble more slowly

  • The discussion (around line 630) poses the idea that divergent HIV-2 CA residues may confer a fitness advantage by evading host innate defense mechanisms. Have the amino acid residues recognized by TRIM5a or MXB been defined by HIV-2? If so, do those binding interfaces align with the CA-CA interfaces?

  • Can the authors speculate if these data provide a mechanistic rationale for why HIV-2 infection/pathogenesis pales in comparison to HIV-1?

Author Response

Comments 1: Cook et al evaluate the assembly kinetics and biochemical characteristics of HIV-2 capsid protomers compared to what it known regarding HIV-1 assembly requirements. While much is known with regard to HIV-1, HIV-2 is much less studied in all capacities. The authors thoroughly and carefully investigate numerous experimental conditions that impact HIV-2 capsid assembly positively or negatively with respect to “normal” conditions established for HIV-1 capsid. Additionally, the authors investigate the functional contribution of divergent HIV-1 and HIV-2 amino acid pairs between CA-CA interfaces and the impact on VLP assembly. Overall, this is a well-written and thorough study with very nice complementary data provided as supplemental material. I have no major concerns regarding this study and only have minor comments/questions.

Response 1: We thank the reviewer for the positive feedback and taking the time to review our manuscript.

Comments 2: Is the HIV-2 capsid sensitive to Lenacapavir?

Response 2: Current data indicate that the HIV-2 capsid is susceptible to lenacapavir, but with reduced potency and a narrow resistance barrier. In vitro, lenacapavir shows low-nanomolar activity against HIV-2, with ~11–14-fold less potent than against HIV-1 (J Infect Dis 2024;229:1290–1294). In vivo, salvage-therapy experience suggests an initial antiviral response followed by rapid emergence of HIV-2 capsid mutations and loss of efficacy within a year (Clin Infect Dis 2024; doi:10.1093/cid/ciae650). Given this, we remain interested in this interaction; however, it lies beyond the scope of this manuscript and will be addressed in future studies.

Comments 3: It would be interesting to determine if the HIV-2 CA-CA “revertant” mutants that exhibit enhanced capsid assembly kinetics also exhibit increase infection efficiency compared to WT or mutants that assemble more slowly

Response 3: We agree that this would be an exciting direction of inquiry and would provide another means of querying the role of capsid stability in infection process of either HIV-1 or HIV-2. However, our group currently lacks the necessary materials, particularly for conducting HIV-2 infectivity assays.

Comments 4: The discussion (around line 630) poses the idea that divergent HIV-2 CA residues may confer a fitness advantage by evading host innate defense mechanisms. Have the amino acid residues recognized by TRIM5a or MXB been defined by HIV-2? If so, do those binding interfaces align with the CA-CA interfaces?

Response 4: We thank the reviewer for this thoughtful question. Most evidence in the literature supports interactions between the CypA-binding loop of CA and the PRYSPRY domain of TRIM5α. In HIV-2, exchanging this loop or specific residues with those from TRIM-resistant SIV strains of Old World monkeys correlated with increased infectivity when challenged by OWM TRIMs. Complicating this, mixed evidence has been presented regarding the roles of three potential Pro residues (positions 120, 160, and 179), all of which are positioned at or near surface loops and distant from the CypA-binding loop, although position 120 lies within a loop which also interacts with CypA in the HIV-1 context. Considering these surfaces collectively, they are all distal to the CA-CA interfaces. Notably, a study we cite in the discussion reported increased TRIM restriction sensitivity in HIV-1 CA upon reversion of Q50 to Tyr, the same interface residue (Y50) which we explored here in HIV-2. The paper characterizes structural differences in regard to the N-terminal β-hairpin and perhaps to the flexibility of the CypA-binding loop, implying that even residues not directly part of the CA–CA interfaces could exert propagating or allosteric effects that alter TRIM5α sensitivity.

Regarding MxB, there is less clear evidence regarding interaction with the HIV-2 capsid. HIV-2 appears less sensitive to MxB restriction than HIV-1. The best-characterized HIV-1 interface involves the MxB C-terminal RRR motif and a highly acidic patch at the CA-CTD trimer interface - residues that are conserved in HIV-2 (though the cytoplasm-exposed dimer interface is less negatively charged in HIV-2). In HIV-1, CypA and/or Nup358 binding to the capsid may protect against MxB restriction, whereas in HIV-2, CypA binds weakly and has only a modest effect on infection. Some other residues have been studied with respect to MxB/HIV-1 interaction, mostly around the FG-binding pocket, but these are also largely conserved between HIV-1 and HIV-2. Altogether, determinants of MxB restriction in HIV-2 remain incompletely defined, particularly given the complex interplay with CypA and Nup358.

Comments 5: Can the authors speculate if these data provide a mechanistic rationale for why HIV-2 infection/pathogenesis pales in comparison to HIV-1?

Response 5: We appreciate the reviewer’s thoughtful question. The field has experienced a significant expansion of evidence across a variety of modalities regarding HIV-1 capsid stability and fate. As we mention in the manuscript, it appears clear from these efforts that capsid stability is carefully tuned to the goals of viral infection, thus we believe our data add to this ongoing conversation. However, while we believe that our results point to intriguing differences in the biochemical characteristics of CA assembly between HIV-1 and HIV-2, we are leery of speculating too far into the implications of these differences for such a multifaceted system. To test the hypothesis that HIV-2 capsid is less stable and therefore more prone to premature disassembly, for example, the in-vitro approaches we describe would be informative in combination with carefully designed in-cell/in-vivo experiments that disentangle intrinsic CA effects from host-factor interactions and internal pressure during reverse-transcription progression - for example, synchronized single-cycle infections with real-time uncoating readouts, nuclear-import assays, and targeted perturbation of CypA/Nup358 and restriction factors (TRIM5α, MxB). These future endeavors may shed more light on the attenuated infection/pathogenesis of HIV-2.